# Impact of Sex on Rehospitalization Rates and Mortality of Patients with Heart Failure with Preserved Ejection Fraction: Differences Between an Analysis Stratified by Sex and a Global Analysis

**DOI:** 10.3390/jpm15070297

**Published:** 2025-07-08

**Authors:** Victoria Cendrós, Mar Domingo, Elena Navas, Miguel Ángel Muñoz, Antoni Bayés-Genís, José María Verdú-Rotellar

**Affiliations:** 1Department of Medicine, Universitat de Barcelona, 08036 Barcelona, Spain; vcendros.bcn.ics@gencat.cat; 2Heart Failure Unit and Cardiology Department, Hospital Universitari Germans Trias i Pujol, 08916 Badalona, Spain; madote@gmail.com (M.D.); abayesgenis@gmail.com (A.B.-G.); 3Atenció Primària Barcelona Ciutat, Institut Català de la Salut—Fundació Institut Universitari per a la Recerca a l’Atenció Primària de Salut Jordi Gol i Gurina (IDIAPJGol), Departament de Salut, Generalitat de Catalunya, Numància, 23, 4a Planta, 08029 Barcelona, Spain; enavas@idiapjgol.info (E.N.); mamunoz.bcn.ics@gencat.cat (M.Á.M.)

**Keywords:** ejection fraction, sex, heart failure, prognosis

## Abstract

**Background:** Differences in the prognosis and associated factors in patients with heart failure with a preserved fraction (HFpEF) according to sex remain uncertain. **Objective:** The objective was to determine the relevance of sex-stratified predictive models in determining prognosis in HFpEF patients. **Methods:** The study was a retrospective, multicenter study of patients previously hospitalized with ejection fraction ≥ 50% (HFpEF) using data from the SIDIAP database. The endpoints were mortality and rehospitalization. Predictive models were performed. **Results:** We identified 2895 patients with HFpEF who were 57% female, with a mean age of 77 (standard deviation [SD] 9.7) years and a median follow-up of 2.0 (IQR 1.0–9.0) years. In the overall analysis, male sex was associated with a higher risk of mortality (HR 1.26, 95% CI 1.06–1.49, *p* = 0.008) and rehospitalization (HR 1.14, 95% CI 1.03–1.33, *p* = 0.04). After sex stratification, the mortality rates per 1000 patient years were 10.40 (95% CI 9.34–11.46) in men and 10.21 (95% CI 9.30–11.11) in women (*p* = 0.7), and the rehospitalization rates were 17.11 (95% CI 16.63–18.58) in men and 17.29 (95% CI 16.01–18.57) in women (*p* = 0.23). In men, the factors related to mortality were age (hazard ratio [HR] 3.14, 95% confidence interval [CI] 2.43–4.06), and hemoglobin (0.84, 0.79–0.89), while in women, they were age (HR 2.92, 95% CI 2.17–3.92), BMI < 30 kg/m^2^ (1.7, 1.37–2.11), diuretics (1.46, 1.11–1.94), and a Charlson > 2 (1.86, 1.02–3.38). Rehospitalization in men was associated with age (HR 1.58, 95% CI 1.23–2.02), BMI < 30 kg/m^2^ (0.75, 0.58–0.95), atrial fibrillation (1.36, 1.07–1.73), hemoglobin (0.91, 0.87–0.95), and coronary disease (1.35, 1.01–1.81). In women, the factors were age (HR 1.33, 95% CI 1.0–1.64), atrial fibrillation (1.57, 1.30–1.91), hemoglobin (0.86, 0.80–0.92), and diuretics (1.37, 1.08–1.73). **Conclusions:** Non-stratified analyses underestimate the poor prognosis in women with HFpEF. Future studies should include analyses stratified by sex.

## 1. Introduction

Heart failure (HF) is a common condition associated with high morbidity and mortality, placing a considerable burden on patients, caregivers, and healthcare systems [1]. Despite a similar incidence between the sexes, diagnosis in women tends to occur at more advanced stages, with corresponding negative implications for recovery and survival [2].

HF can be classified into reduced ejection fraction (HFrEF) and preserved ejection fraction (HFpEF), and there is emerging evidence that there may be epidemiological, pathophysiological, clinical, and treatment-related differences between the sexes in relation to the ejection fraction category. Recent research has highlighted sex-related differences in HFpEF pathophysiology. Women show enhanced immune activation and a greater tendency toward microvascular dysfunction, which may contribute to disease development. Differences in hormonal regulation and biomarker expression have also been observed, suggesting distinct biological mechanisms between sexes that may influence prognosis and response to therapy [2]. Recent research has highlighted sex-related differences in HFpEF pathophysiology. Women show enhanced immune activation and a greater tendency toward microvascular dysfunction, which may contribute to disease development. Differences in hormonal regulation and biomarker expression have also been observed, suggesting distinct biological mechanisms between the sexes that may influence prognosis and response to therapy. Despite this, the current guidelines for the diagnosis and treatment of HF remain largely uniform for both men and women [2]. However, this omission may be due to the fact that the sex-related differences and the implications for prognosis are still uncertain.

Data suggest that the comorbidities that affect the type and prognosis of heart failure differ depending on sex. Atrial fibrillation, obesity, diabetes, hypertension, and anemia are more prevalent in women [2,3,4]. In contrast, men present a higher prevalence of smoking and chronic obstructive pulmonary disease (COPD), but these conditions have been shown to result in worse HF outcomes when present in women [2,3,4]. Regarding ejection fraction, HFrEF is more common in men, primarily due to a higher incidence of ischemic heart disease in a large proportion of cases. In women, HFpEF is more frequent, with the predominant risk factors including hypertension, diabetes, obesity, and autoimmune diseases. These factors promote a systemic inflammatory response that plays a key role in endothelial dysfunction and structural changes in the myocardium [2,3,4,5].

Current clinical practice guidelines do not differentiate pharmacological management by sex. However, differences in the prescription of certain medications have been reported, which are attributed to the higher prevalence of ischemic heart disease in men [2]. Additionally, social inequalities, which influence both lifestyle and access to certain treatments, as well as a higher rate of adverse effects and treatment discontinuation, have been noted to affect women more frequently [2]. Evidence suggests that women may achieve therapeutic goals with lower doses than those recommended in the guidelines [5] and that they could benefit from certain treatments (valsartan–sacubitril, spironolactone, SGLT2 inhibitors) at higher ejection fractions than men [3,4,5,6]. 

Overall, the evidence suggests that there may be significant sex-related differences related to the diagnosis, type, and treatment of heart failure, which may differentially affect the prognosis. However, various reviews estimate that women account for only approximately 20–30% of participants in heart failure studies, despite representing about half of the population affected by the disease [1,7,8].

Most published studies show a higher probability of mortality and hospitalizations in men than in women with HFpEF [1,3,9,10,11]. However, these studies are based on the analysis of the entire population, considering sex as one more variable in a multivariate analysis, without taking into account differences due to the sex of the participants. Recently, our group has also published a paper on the subject, in which we performed an analysis of the global population [12]. Recent reviews [7,8] recommend routinely conducting and planning statistical analyses stratified by sex. So we wondered if, when performing the stratified analysis by sex, the variables and their relationship with morbidity and mortality would be different.

The objective of our study was to determine the relevance of sex-stratified predictive models in HFpEF patients for evaluating sex-specific differences in prognosis, rehospitalization, and mortality rates.

## 2. Materials and Methods

### 2.1. Study Design and Data Source

We conducted a retrospective study using data from the Information System for Research in Primary Care (SIDIAP), a population-based database managed by the Institut Català de la Salut.

SIDIAP includes anonymized, standardized clinical records of approximately 8 million individuals—representing around 80% of the Catalan population and 10% of the Spanish population—collected since 2006 across 328 primary care centers [13].

The validity and representativeness of the database have been established in previous research, particularly in studies on cardiovascular conditions and related risk factors [14].

The available data encompass demographics, diagnoses (ICD-10), laboratory results, socio-economic variables, vaccination history, medication dispensation from community pharmacies, and mortality.

In addition, SIDIAP can be linked to external sources, such as hospital discharge registries coded under ICD-9, for more comprehensive clinical follow-up. SIDIAP has access to all hospital discharges that have taken place in the public network of the Catalan Institute of Health hospitals (SIDIAP-H) since 2006 (30% of the SIDIAP population); this register includes diagnoses and hospital procedures.

The study was performed in accordance with the Guidelines of the Helsinki Declaration and was approved by the Clinical Research Ethics Committee of the IDIAPJGol (P18/010).

### 2.2. Patients

We included all individuals aged over 18 years with a documented left ventricular ejection fraction (LVEF) ≥ 50% recorded in the SIDIAP database, with a HF discharge diagnosis in any position from the SIDIAP-H. LVEF ≥ 50% was chosen because it is the cut-off point for the diagnosis of HFpEF recommended by the European Society of Cardiology clinical guideline [5]. When echocardiographic measurements and NT-proBNP levels were unavailable, a clearly recorded hospital admission for heart failure was accepted as evidence of a confirmed diagnosis [15].

### 2.3. Clinical Variables

Clinical information was extracted from the SIDIAP database. Individuals were classified as non-smokers if they had never smoked or had smoking cessation >1 year. Obesity was defined by a body mass index (BMI) of ≥30 kg/m^2^. Chronic kidney disease was identified by an estimated glomerular filtration rate (eGFR) below 60 mL/min/1.73 m^2^. Anemia was defined following the WHO criteria: hemoglobin levels < 13 g/dL in men and <12 g/dL in women. The Charlson comorbidity index was used to quantify the comorbidity burden and estimate the mortality risk.

### 2.4. Follow-Up and Events

Patients were enrolled from 2009 to 2017, with follow-up extending through December 2018. The primary endpoints were all-cause death and first HF rehospitalization. Hospitalization for HF was considered as an event when it was the first diagnosis discharge code or the second one when acute respiratory insufficiency was the first hospitalization event. Mortality data were collected from the SIDIAP database.

### 2.5. Statistical Analysis

Continuous variables were described as the mean ± standard deviation (SD) or the median with the interquartile range (IQR), depending on the data distribution. Categorical variables were summarized using absolute frequencies and percentages. Missing values were neither imputed nor replaced.

To assess associations between variables, the Chi-square test was used for categorical comparisons and the Student’s *t*-test for continuous variables. Kaplan–Meier survival curves were generated for the composite endpoint, while cumulative incidence functions were plotted separately for all-cause death and HF-related rehospitalizations.

Event rates (mortality and rehospitalization) were calculated per 1000 person–years for the overall cohort and stratified by sex. Associations between the clinical variables and outcomes were explored using univariable and multivariable Cox proportional hazards models (backward stepwise approach), with an assessment of proportionality and linearity assumptions.

For analyses focusing on HF rehospitalization, where death could act as a competing event, the Fine and Gray subdistribution hazard model was applied. Prognostic models for both mortality and rehospitalization were developed first for the entire sample and then separately by sex, including covariates with *p*-values < 0.05 in multivariable analysis. All statistical analyses were performed using R software (version 3.6.1; Vienna, Austria).

## 3. Results

### 3.1. Demographics and Clinical Data of Study Patients

Among the 37,822 patients identified with a heart failure discharge diagnosis in SIDIAP-H, 32,263 lacked recorded LVEF data, 2664 had an LVEF < 50%, and 2895 patients with an LVEF ≥ 50% were included in the final analysis. Only 1% of the participants were lost to follow-up.

Table 1 and Table 2 show the baseline demographic, clinical characteristics, and treatments of the study population by sex and according to mortality and heart failure readmission event, respectively.

In summary, of the 2898 patients included, 57.4% were women. Compared to men, women with HFpEF were significantly older and presented a higher proportion of obesity, hypertension, hyperlipidemia, atrial fibrillation, and CKD. In addition, fewer women were smokers, and fewer had diabetes, coronary and peripheral arterial disease, and COPD. Women also had a lower comorbidity according to the Charlson index. Regarding treatments, a greater proportion of women were treated with diuretics and digoxin, and twice as few women with ischemic heart disease were treated with statins compared to men.

### 3.2. Mortality

In the overall analysis, the variables independently associated with mortality in the Cox regression model were male sex, older age, BMI < 30 kg/m^2^, Charlson comorbidity index ≥ 3, and use of loop diuretics. Higher hemoglobin levels were linked to lower mortality risk. When stratifying the model according to sex, we found that, for men, only age and hemoglobin levels were related to mortality. In women, age, BMI < 30 kg/m^2^, use of loop diuretics, and the Charson index were directly related to mortality, while the LVEF ≥ 55% was protective.

During a median follow-up of 2.0 [IQR 1.0–9.0] years, 864 (29.8%) patients died. Mortality per 1000 patient–years was 10.29 (95% CI 9.6–10.9). In the multivariate analysis for the entire population, sex was an independent predictor of mortality: the mortality rate was 26% higher for men than women. The forest plot of multivariable Cox regression analyses for all-cause mortality is stratified by sex. Hazard ratios (HR) and 95% confidence intervals (CI) are shown in Figure 1.

However, when the analysis was stratified by sex, no significant differences in mortality were observed (*p* = 0.7) (Figure 2). The mortality rate per 1000 patient–years in men was 10.40 (95% CI 9.34–11.46) versus 10.21 (95% CI 9.30–11.11) in women.

### 3.3. Heart Failure Rehospitalization

In the multivariable analysis of the overall cohort, the factors independently associated with HF-related readmission included male sex, older age, presence of atrial fibrillation, and use of loop diuretics. Higher hemoglobin levels were associated with a reduced risk of hospitalization. When stratifying the model according to sex, the variables related to the probability of being hospitalized were different depending on the sex of patients. In men, age, BMI < 30 kg/m^2^, the presence of atrial fibrillation, and ischemic heart disease were associated with rehospitalization, while in women, the variables were age, atrial fibrillation, and treatment with loop diuretics (Figure 3).

During the study period, 831 (40.3%) patients were rehospitalized. The HF rehospitalization rate per 1000 patient–years was 17.21 (95% CI 16.24–18.18).

In the multivariate analysis for the entire population, sex was an independent predictor for rehospitalization. Men had a rehospitalization rate 14% higher than women. The forest plot of multivariable Cox regression analyses for HF-related rehospitalization is stratified by sex. Hazard ratios (HR) and 95% confidence intervals (CI) are shown in Figure 3.

However, when the analysis was stratified by sex, no significant differences in the rehospitalization rates were observed (*p* = 0.23) (Figure 4).

The rehospitalization rate per 1000 patient–years in men was 17.11 (95% CI 16.63–18.58) versus 17.29 (95% CI 16.01–18.57) in women.

Appendix A includes a detailed list of the dichotomous independent variables used in the multivariable Cox regression models.

## 4. Discussion

We have shown that, in a multivariate analysis of the entire population, mortality and rehospitalization rates were higher for men than women. However, a sex-stratified analysis revealed no statistically significant differences between the groups. Therefore, the apparent “protective effect” of female sex was lost. The present study, therefore, shows that the analysis of the outcomes in heart failure patients without stratifying by sex can underestimate the probability of death and rehospitalization in women with HFpEF. When performing the stratified analysis by sex, we also observed that the variables related to these outcomes were different in women.

In our cohort, female patients with HFpEF were older than their male counterparts, which is consistent with the findings reported in the previous literature [1,2,3,4,5,9,11]. Several studies suggest that the pathophysiology, prevalence, and influence of different risk factors on the pathogenesis of HF are different between the sexes [1,2,4,9,16]. The ischemic etiology that is fundamental in men is less frequent in women. In contrast, HFpEF in women may be fundamentally related to risk factors such as obesity, hypertension, atrial fibrillation, and diabetes, which may promote an increased inflammatory status that causes myocardial dysfunction [2,4,14,16].

In our study, we confirmed the higher prevalence of these risk factors in women, except for the prevalence of diabetes. In this respect, the association between the sex of the patients and the presence of diabetes remains uncertain, with contradictory results in the literature [2,11]. In our study, women had a significantly higher ejection fraction than men, which is in line with previous publications [17,18]. Myocardial remodeling differs between men and women with HF. Men tend to have more eccentric hypertrophy, and women tend to have smaller cardiac chambers with a higher LVEF, which can cause diastolic dysfunction with a higher LVEF [19,20]. Recent evidence has questioned whether the classification thresholds for LVEF should be the same for men and women. A subanalysis of the PARAGON-HF trial [21] suggested that the prognostic cut-off for LVEF may be higher in women. However, in the same cohort, Kondo et al. did not find statistically significant sex-based differences in the outcomes [22]. Interestingly, while some studies, such as Wehner et al. [23], report worse outcomes at higher LVEF values (i.e., supranormal EF), others show no increased risk in women with an LVEF > 60%. Overall, there appears to be no clear linear relationship between LVEF and prognosis when the EF is above 40%. In our study, an LVEF > 55% was associated with a protective effect in women. Further research is needed to clarify the implications of supranormal EF on outcomes, particularly through sex-stratified analyses, as the current findings remain inconclusive.

The under-treatment with statins and antiplatelet agents in women with coronary heart disease, which has also been found in other published studies, is concerning [2]. In addition, the higher proportion of women treated with digoxin is also remarkable, given the potential higher risk of mortality and poor safety profile of this treatment in women compared to men. In this study, a higher proportion of women than men were treated with loop diuretics. All of these findings are in line with other studies reporting lower adherence to guideline recommendations and a higher proportion of symptomatic treatment in women [2].

Subgroup analyses from recent clinical trials have highlighted potential sex differences in treatment response among patients with HFpEF. In the PARAGON-HF trial, sacubitril–valsartan appeared to reduce HF hospitalizations more effectively in women, particularly those with LVEF between 45% and 57% [21]. Similarly, in the TOPCAT trial, women had a higher mean LVEF than men and experienced significantly lower mortality [24]. These findings suggest that sex-specific responses may influence treatment efficacy in HFpEF. In our study, no specific treatment showed a statistically significant association with outcomes, possibly reflecting differences in the baseline characteristics or treatment allocation in routine clinical practice.

Whether there are differences in the prognosis of HFpEF depending on the sex of patients remains uncertain. Women may be at lower risk compared with men when considering longer-term outcomes. In the I-PRESERVE trial [10], female patients hospitalized with HFpEF exhibited a 20% lower risk of death from both cardiovascular and non-cardiovascular causes. Similarly, the MAGGIC meta-analysis reported better survival outcomes in women [9].

The study by Gracia et al. found the probability of death was 30% greater in men than in women, as well as a higher frequency of hospital readmissions [11]. However, other studies, such as the one published by Chung et al. [25] and the study based on the Italian network on congestive HF registry [26], showed that sex was not an independent predictor of mortality in patients with HFpEF. Of note, most of the aforementioned studies were based on the entire population and considered the sex of patients as one variable in the multivariate analyses. In our study, if we only considered the results of the sample as a whole, we would obtain a probability greater than 26% for mortality and 14% for readmission in men. However, when performing the analysis stratified by sex, we found that, in the case of mortality, these differences did not exist. Therefore, our study suggests it may be important to include analyses stratified by sex in other studies to determine if this important result is consistent across other cohorts from different regions.

Furthermore, in the analysis stratified by sex, we found that the variables related to the outcomes did not agree with the global model. In the case of mortality, in men, only hemoglobin levels and older age retained significance, while in women, the model was very similar to the global one, but an LVEF ≥ 55% was protective. This may be attributed to sex-related differences in both pathogenesis and the impact of specific risk factors. The pro-inflammatory state in women and the history of ischemic heart disease in men play a crucial role. In fact, several studies have shown a worse prognosis for the morbidity and mortality of these risk factors in women [2,15], and recently, in a study about the influence of phenotypes in HFpEF, Iorio defined a diverse cluster [27], with females with multiple cardiovascular risk factors having the poorest prognosis.

### 4.1. Study Limitations

The retrospective nature of the data collection from public primary care centers and hospitals may have introduced selection and information biases.

These results are derived from a specific ethnodemographic cohort (the Catalan population) and may not be directly generalizable to populations with different ethnic compositions or health systems. Caution should be taken when extrapolating to more heterogeneous or internationally diverse populations.

Further, we only had data available for the global mortality, and we could not obtain a list with the different causes.

Additionally, the definition of HFpEF based solely on an LVEF ≥ 50% without the availability of natriuretic peptides or diastolic function measures may have led to diagnostic misclassification. To reduce this risk, we included only patients with a documented prior hospital admission for heart failure, which has been shown to enhance diagnostic reliability in EHR-based studies [15].

### 4.2. Implications and Future Lines of Research

As indicated by Lala et al. in their review [2], strategies to increase the representation of women in clinical studies are crucial. Studies should be oriented towards the differentiation in the analysis of data, stratifying the outcomes by the sex of the participants.

## 5. Conclusions

Non-sex-stratified analyses in patients with HFpEF may underestimate the risk in women. Sex-stratified models could support improved clinical decision-making and adjustment of diagnostic, follow-up, and treatment protocols.

## Figures and Tables

**Figure 1 jpm-15-00297-f001:**
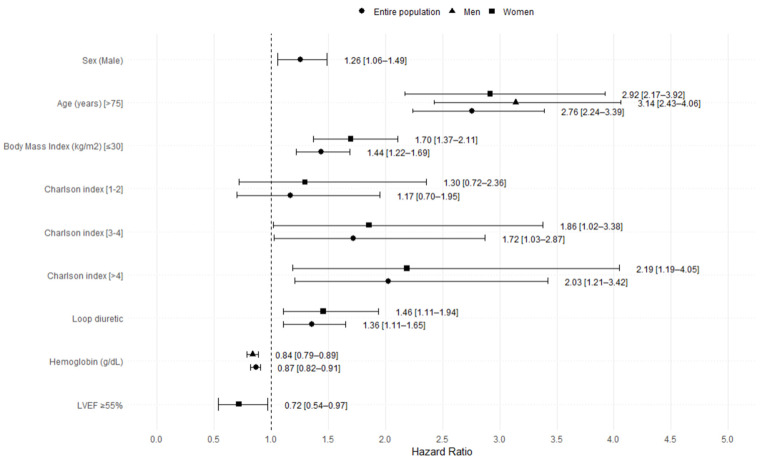
Forest plot of multivariable Cox regression analyses for all-cause mortality, stratified by sex.

**Figure 2 jpm-15-00297-f002:**
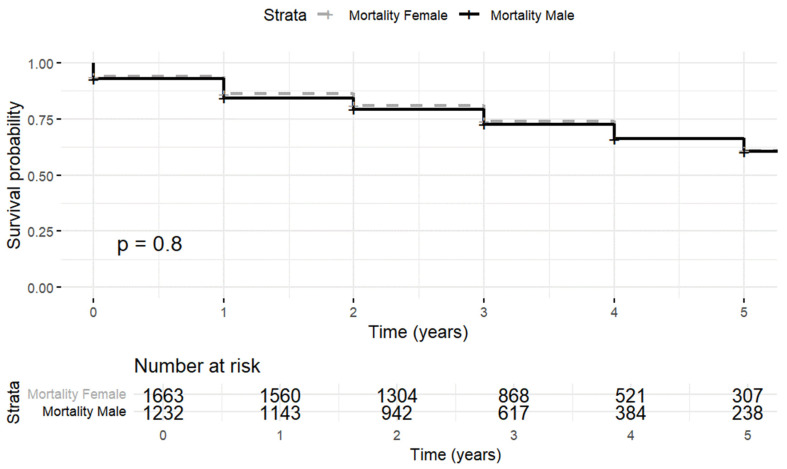
Kaplan–Meier curves of all-cause mortality by sex.

**Figure 3 jpm-15-00297-f003:**
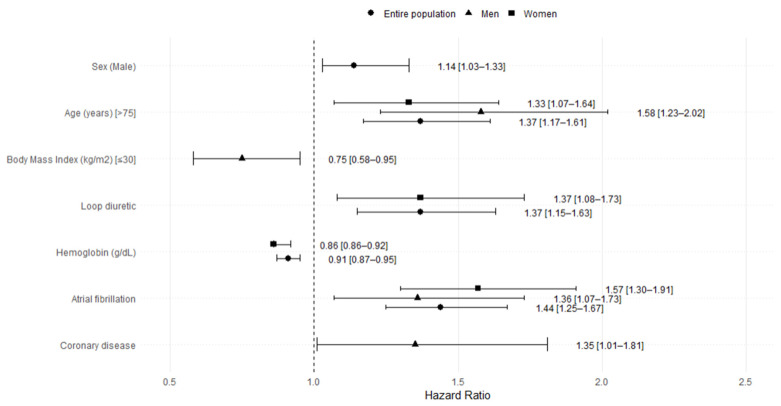
Forest plot of multivariable Cox regression analyses for heart failure-related rehospitalization, stratified by sex.

**Figure 4 jpm-15-00297-f004:**
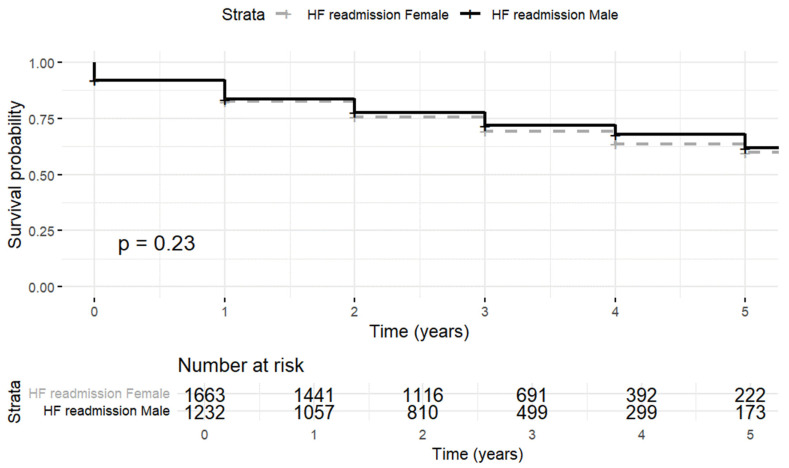
Kaplan–Meier curves of rehospitalization by sex.

**Table 1 jpm-15-00297-t001:** Baseline characteristics of the study population by sex according to mortality event.

	NO Mortality Event				Mortality Event			
	[ALL] (N = 2031)	Female (N = 1170)	Male (N = 861)	*p*-Value	[ALL] (N = 864)	Female (N = 493)	Male (N = 371)	*p*-Value
Age (years), mean (SD)	75.4 (9.9)	77.4 (8.9)	72.7 (10.6)	<0.001	80.9 (8.1)	81.7 (7.5)	79.8 (8.8)	0.001
Tobacco, N (%)				<0.001				<0.001
Non-smoker	1834 (91.8%)	1118 (97.0%)	716 (84.8%)		810 (95.1%)	472 (97.9%)	338 (91.4%)	
Smoker	163 (8.1%)	35 (3.0%)	128 (15.2%)		42 (4.9%)	10 (2.0%)	32 (8.6%)	
Clinical variables								
Body mass index, mean (SD)	31.6 (6.0)	32.3 (6.4)	30.6 (5.4)	<0.001	29.9 (6.0)	30.5 (6.7)	29.1 (4.8)	0.001
^a^ Obesity, N (%)	906 (54.8%)	566 (60.0%)	340 (48.0%)	<0.001	289 (43.7%)	173 (47.0%)	116 (39.6%)	0.067
Hypertension, N (%)	1648 (81.1%)	981 (83.8%)	667 (77.5%)	<0.001	706 (81.7%)	419 (85.0%)	287 (77.4%)	0.005
Diabetes Mellitus, N (%)	905 (44.6%)	486 (41.5%)	419 (48.7%)	0.002	397 (45.9%)	227 (46.0%)	170 (45.8%)	1.000
Microalbuminuria, N (%)	406 (20.0%)	206 (17.6%)	200 (23.2%)	0.002	183 (21.2%)	93 (18.9%)	90 (24.3%)	0.066
Dyslipidemia, N (%)	1067 (52.5%)	630 (53.8%)	437 (50.8%)	0.182	431 (49.9%)	257 (52.1%)	174 (46.9%)	0.146
Coronary artery disease, N (%)	314 (15.5%)	132 (11.3%)	182 (21.1%)	<0.001	159 (18.4%)	75 (15.2%)	84 (22.6%)	0.007
Valvular heart disease, N (%)	587 (28.9%)	368 (31.5%)	219 (25.4%)	0.004	279 (32.3%)	169 (34.3%)	110 (29.6%)	0.172
Atrial fibrillation, N (%)	974 (48.0%)	600 (51.3%)	374 (43.4%)	0.001	480 (55.6%)	279 (56.6%)	201 (54.2%)	0.524
Stroke, N (%)	210 (10.3%)	119 (10.2%)	91 (10.6%)	0.828	102 (11.8%)	60 (12.2%)	42 (11.3%)	0.782
Peripheral artery disease, N (%)	165 (8.1%)	51 (4.3%)	114 (13.2%)	<0.001	95 (11.0%)	30 (6.0%)	65 (17.5%)	<0.001
^b^ Anemia, N (%)	326 (17.2%)	168 (15.2%)	158 (20.2%)	0.006	261 (32.8%)	131 (29.0%)	130 (37.8%)	0.011
^c^ Chronic kidney disease, N (%)	548 (27.0%)	345 (29.5%)	203 (23.6%)	0.004	358 (41.4%)	215 (43.6%)	143 (38.5%)	0.154
COPD, N (%)	626 (30.8%)	310 (26.5%)	316 (36.7%)	<0.001	317 (36.7%)	135 (27.4%)	182 (49.1%)	<0.001
Charlson index, N (%)				0.001				<0.001
0	139 (6.8%)	88 (7.5%)	51 (5.9%)		30 (3.4%)	26 (5.2%)	4 (1.0%)	
(1,2)	860 (42.3%)	525 (44.9%)	335 (38.9%)		261 (30.2%)	165 (33.5%)	96 (25.9%)	
(3,5)	688 (33.9%)	389 (33.2%)	299 (34.7%)		350 (40.5%)	197 (40.0%)	153 (41.2%)	
>4	344 (16.9%)	168 (14.4%)	176 (20.4%)		223 (25.8%)	105 (21.3%)	118 (31.8%)	
LVFE, mean (SD)	61.7 (7.3)	62.5 (7.3)	60.6 (7.1)	<0.001	61.0 (7.2)	61.6 (7.0)	60.3 (7.4)	0.015
LVFE, N (%)				<0.001				0.025
≥55%	1726 (85.0%)	1035 (88.5%)	691 (80.3%)		708 (81.9%)	417 (84.6%)	291 (78.4%)	
<55%	305 (15.0%)	135 (11.5%)	170 (19.7%)		156 (18.1%)	76 (15.4%)	80 (21.6%)	
Treatment variables								
ACEi/ARB, N (%)	1431 (70.5%)	839 (71.7%)	592 (68.8%)	0.164	576 (66.7%)	340 (69.0%)	236 (63.6%)	0.114
Beta-blockers, N (%)	1042 (51.3%)	601 (51.4%)	441 (51.2%)	0.983	383 (44.3%)	228 (46.2%)	155 (41.8%)	0.215
MRA, N (%)	198 (9.7%)	113 (9.6%)	85 (9.8%)	0.932	79 (9.1%)	43 (8.7%)	36 (9.7%)	0.707
Loop diuretic, N (%)	1395 (68.7%)	843 (72.1%)	552 (64.1%)	<0.001	670 (77.5%)	400 (81.1%)	270 (72.8%)	0.005
Tiazide, N (%)	186 (9.1%)	115 (9.8%)	71 (8.2%)	0.252	84 (9.7%)	51 (10.3%)	33 (8.8%)	0.551
Digoxin, N (%)	268 (13.2%)	182 (15.6%)	86 (9.9%)	<0.001	138 (16.0%)	98 (19.9%)	40 (10.8%)	<0.001
Calcium channel blockers, N (%)	617 (30.4%)	346 (29.6%)	271 (31.5%)	0.383	272 (31.5%)	161 (32.7%)	111 (29.9%)	0.433
Anti-platelet drugs, N (%)	736 (36.2%)	357 (30.5%)	379 (44.0%)	<0.001	344 (39.8%)	199 (40.4%)	145 (39.1%)	0.756
Anticoagulants, N (%)	812 (40.0%)	480 (41.0%)	332 (38.6%)	0.282	345 (39.9%)	200 (40.6%)	145 (39.1%)	0.711
Statins, N (%)	1071 (52.7%)	576 (49.2%)	495 (57.5%)	<0.001	388 (44.9%)	228 (46.2%)	160 (43.1%)	0.399
Laboratory variables								
Hemoglobin, mean (SD)	13.7 (1.6)	13.2 (1.3)	14.4 (1.7)	<0.001	13.1 (1.6)	12.8 (1.5)	13.5 (1.7)	<0.001
Estimated GFR, mean (SD)	70.2 (19.0)	69.1 (18.6)	71.8 (19.4)	0.002	62.6 (20.2)	61.9 (19.7)	63.5 (20.8)	0.272
Serum potassium, mean (SD)	4.7 (0.5)	4.6 (0.5)	4.7 (0.5)	0.003	4.7 (0.5)	4.7 (0.5)	4.8 (0.5)	0.110

ACEi, angiotensin-converting enzyme inhibitors; ARB, angiotensin II receptor blockers; COPD, chronic obstructive pulmonary disease; LVEF, left ventricular ejection fraction; MRA, mineral corticoid receptor antagonist. ^a^ Body mass index > 30 kg/m^2^. ^b^ According to WHO criteria: <13 g/dL in men and <12 g/dL in women. ^c^ Estimated glomerular filtration rate (Chronic Kidney Disease Epidemiology Collaboration equation) < 60 mL/min per 1.73 m^2^.

**Table 2 jpm-15-00297-t002:** Baseline characteristics of the study population by sex according to heart failure readmission event.

	NO Heart Failure Readmission				Heart Failure Readmission			
	[ALL] (N = 2064)	Female (N = 1169)	Male (N = 895)	*p*-Value	[ALL] (N = 831)	Female (N = 494)	Male (N = 337)	*p*-Value
Age (years), mean (SD)	76.7 (10.1)	78.5 (9.1)	74.4 (10.9)	<0.001	77.9 (8.7)	79.2 (7.9)	76.1 (9.5)	<0.001
Tobacco, N (%)				<0.001				<0.001
Non-smoker	1869 (92.0%)	1112 (96.8%)	757 (85.8%)		775 (94.7%)	478 (98.4%)	297 (89.5%)	
Smoker	162 (7.9%)	37 (3.2%)	125 (14.2%)		43 (5.2%)	8 (1.6%)	35 (10.5%)	
Clinical variables								
Body mass index, mean (SD)	30.9 (6.1)	31.8 (6.6)	29.8 (5.2)	<0.001	31.5 (6.0)	31.9 (6.4)	30.9 (5.2)	0.033
^a^ Obesity, N (%)	834 (50.7%)	519 (56.4%)	315 (43.4%)	<0.001	361 (54.0%)	220 (56.1%)	141 (51.1%)	0.227
Hypertension, N (%)	1671 (81.0%)	979 (83.7%)	692 (77.3%)	<0.001	683 (82.2%)	421 (85.2%)	262 (77.7%)	0.007
Diabetes Mellitus, N (%)	858 (41.6%)	454 (38.8%)	404 (45.1%)	0.005	444 (53.4%)	259 (52.4%)	185 (54.9%)	0.529
Microalbuminuria, N (%)	390 (18.9%)	191 (16.3%)	199 (22.2%)	0.001	199 (23.9%)	108 (21.9%)	91 (27.0%)	0.105
Dyslipidemia, N (%)	1056 (51.2%)	608 (52.0%)	448 (50.1%)	0.403	442 (53.2%)	279 (56.5%)	163 (48.4%)	0.026
Coronary artery disease, N (%)	323 (15.6%)	134 (11.5%)	189 (21.1%)	<0.001	150 (18.1%)	73 (14.8%)	77 (22.8%)	0.004
Valvular heart disease, N (%)	613 (29.7%)	364 (31.1%)	249 (27.8%)	0.113	253 (30.4%)	173 (35.0%)	80 (23.7%)	0.001
Atrial fibrillation, N (%)	974 (47.2%)	577 (49.4%)	397 (44.4%)	0.027	480 (57.8%)	302 (61.1%)	178 (52.8%)	0.021
Stroke, N (%)	225 (10.9%)	128 (10.9%)	97 (10.8%)	0.993	87 (10.5%)	51 (10.3%)	36 (10.7%)	0.960
Peripheral artery disease, N (%)	186 (9.0%)	49 (4.1%)	137 (15.3%)	<0.001	74 (8.9%)	32 (6.4%)	42 (12.5%)	0.004
^b^ Anemia, N (%)	379 (19.8%)	183 (16.6%)	196 (24.0%)	<0.001	208 (27.0%)	116 (25.4%)	92 (29.4%)	0.258
^c^ Chronic kidney disease, N (%)	623 (30.2%)	379 (32.4%)	244 (27.3%)	0.013	283 (34.1%)	181 (36.6%)	102 (30.3%)	0.067
COPD, N (%)	661 (32.0%)	305 (26.1%)	356 (39.8%)	<0.001	282 (33.9%)	140 (28.3%)	142 (42.1%)	<0.001
Charlson index, N (%)				<0.001				0.020
0	130 (6.3%)	87 (7.4%)	43 (4.8%)		39 (4.6%)	27 (5.4%)	12 (3.5%)	
[1,2]	817 (39.6%)	514 (44.0%)	303 (33.9%)		304 (36.6%)	176 (35.6%)	128 (38.0%)	
[3,5]	729 (35.3%)	387 (33.1%)	342 (38.2%)		309 (37.2%)	199 (40.3%)	110 (32.6%)	
>4	388 (18.8%)	181 (15.5%)	207 (23.1%)		179 (21.5%)	92 (18.6%)	87 (25.8%)	
LVFE, mean (SD)	61.6 (7.2)	62.3 (7.2)	60.7 (7.1)	<0.001	61.2 (7.3)	61.9 (7.3)	60.1 (7.3)	0.001
LVFE, N (%)				<0.001				<0.001
≥55%	1746 (84.6%)	1023 (87.5%)	723 (80.8%)		688 (82.8%)	429 (86.8%)	259 (76.9%)	
<55%	318 (15.4%)	146 (12.5%)	172 (19.2%)		143 (17.2%)	65 (13.2%)	78 (23.1%)	
Treatment variables								
ACEi/ARB, N (%)	1412 (68.4%)	818 (70.0%)	594 (66.4%)	0.089	595 (71.6%)	361 (73.1%)	234 (69.4%)	0.287
Beta-blockers, N (%)	1006 (48.7%)	581 (49.7%)	425 (47.5%)	0.341	419 (50.4%)	248 (50.2%)	171 (50.7%)	0.935
MRA, N (%)	189 (9.1%)	101 (8.6%)	88 (9.8%)	0.393	88 (10.6%)	55 (11.1%)	33 (9.7%)	0.616
Loop diuretic, N (%)	1414 (68.5%)	840 (71.9%)	574 (64.1%)	<0.001	651 (78.3%)	403 (81.6%)	248 (73.6%)	0.008
Tiazide, N (%)	192 (9.3%)	118 (10.1%)	74 (8.2%)	0.181	78 (9.3%)	48 (9.7%)	30 (8.9%)	0.784
Digoxin, N (%)	261 (12.6%)	185 (15.8%)	76 (8.4%)	<0.001	145 (17.4%)	95 (19.2%)	50 (14.8%)	0.122
Calcium channel blockers, N (%)	606 (29.4%)	337 (28.8%)	269 (30.1%)	0.577	283 (34.1%)	170 (34.4%)	113 (33.5%)	0.850
Anti-platelet drugs, N (%)	760 (36.8%)	378 (32.3%)	382 (42.7%)	<0.001	320 (38.5%)	178 (36.0%)	142 (42.1%)	0.089
Anticoagulants, N (%)	794 (38.5%)	455 (38.9%)	339 (37.9%)	0.661	363 (43.7%)	225 (45.5%)	138 (40.9%)	0.215
Statins, N (%)	1026 (49.7%)	552 (47.2%)	474 (53.0%)	0.011	433 (52.1%)	252 (51.0%)	181 (53.7%)	0.488
Laboratory variables								
Hemoglobin, mean (SD)	13.6 (1.6)	13.2 (1.4)	14.1 (1.8)	<0.001	13.3 (1.7)	12.9 (1.4)	13.9 (1.8)	<0.001
Estimated GFR, mean (SD)	68.4 (19.9)	67.5 (19.3)	69.6 (20.5)	0.021	66.9 (19.1)	65.9 (18.9)	68.4 (19.2)	0.075
Serum potassium, mean (SD)	4.7 (0.5)	4.6 (0.5)	4.7 (0.5)	0.002	4.7 (0.5)	4.7 (0.5)	4.8 (0.6)	0.130

ACEi, angiotensin-converting enzyme inhibitors; ARB, angiotensin II receptor blockers; COPD, chronic obstructive pulmonary disease; LVEF, left ventricular ejection fraction; MRA, mineral corticoid receptor antagonist. ^a^ Body mass index >30 kg/m^2^. ^b^ According to WHO criteria: <13 g/dL in men and <12 g/dL in women. ^c^ Estimated glomerular filtration rate (Chronic Kidney Disease Epidemiology Collaboration equation) < 60 mL/min per 1.73 m^2^.

## Data Availability

The data presented in this study are available only on request from the corresponding author due to restrictions of the Ethics Committee.

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
