# Peer review of "Impact of Sex on Rehospitalization Rates and Mortality of Patients with Heart Failure with Preserved Ejection Fraction: Differences Between an Analysis Stratified by Sex and a Global Analysis"

_jpm, 2025, doi:10.3390/jpm15070297_

Round 1
Reviewer 1 Report
Comments and Suggestions for Authors
My compliments to the authorship group, this is novel data representing the sex-differences in outcomes in the Catalan population in HpEF. A well done paper.
A few suggestions:
- In the limitations please clearly state that the results are applicable to this very specific ethnodemographic cohort and may/or may not be translatable to other more heterogeneous or differing largely homogeneous populations globally.
- May consider presenting your key data from tables in a Hazard ratio/Line of identity plots.
- Dichotomized analysis of clinical outcomes would also be helpful for further hypothesis generation.
Author Response
Response to Reviewer 1
We sincerely thank Reviewer 1 for their positive feedback and constructive suggestions. The following changes were made in response:
- Limitations and generalizability:
We have now explicitly stated in the limitations section that our results are applicable to a specific ethnodemographic cohort (Catalan population with HFpEF) and may not be directly generalizable to more heterogeneous or demographically distinct populations. This point has been added in the revised manuscript
- Hazard ratio/Line of identity plots:
We appreciate the suggestion and have incorporated a supplementary forest plot presenting the key data using hazard ratios to facilitate visual interpretation of the differences. This figure replaces the previous Tables 3 and 4 in the main manuscript. The change has been reflected in the revised version and is referenced in the Results section
- Dichotomized analysis of outcomes:
We agree that a dichotomized analysis could be useful for hypothesis generation. Therefore, we have performed additional multivariable logistic regression analyses using dichotomous outcomes (e.g., mortality: yes/no; rehospitalization: yes/no), stratified by sex, to complement the time-to-event Cox models. These results have been added as Supplementary Material 1, along with a detailed list of the dichotomous independent variables used in the analyses.
Reviewer 2 Report
Comments and Suggestions for Authors
The study tackles a relevant question about sex differences in HFpEF outcomes, but a few points could strengthen it further. The abstract sets up the question well, but the phrasing around the paradox of stratification wiping out observed differences could be clearer, and reporting key numbers would help readers grasp the impact quickly. The introduction is generally solid but feels a bit standard; adding recent mechanistic insights like sex differences in microvascular dysfunction, immune aging, or fibrosis patterns would bring it up to date with where the field is moving. The methods are well described, but defining HFpEF solely by LVEF ≥50% without natriuretic peptides or diastolic function measures risks including patients without true HFpEF, which is a known limitation in EHR studies. Also, relying on backward stepwise regression may limit the robustness of your predictive models; consider discussing why this was chosen over penalized methods, and clarify how missing data were handled, as this can significantly affect results. The results are clearly presented but would benefit from explicitly reporting effect sizes with confidence intervals and providing details on model diagnostics and calibration to support your conclusions. The finding that higher LVEF is protective in women is interesting but should be discussed carefully given emerging evidence of non-linear relationships between EF and outcomes. The discussion aligns with your findings but could dig deeper by connecting to recent trial subgroup analyses on sex differences in HFpEF treatment response, which would increase the paper’s relevance to current debates. The limitations are acknowledged but could more clearly state the potential for residual confounding and the absence of key echo parameters. Finally, the conclusions are reasonable but would benefit from softer phrasing to avoid implying causality and from a clearer explanation of how sex-stratified models could improve clinical decision-making. Overall, this is a meaningful contribution but would be stronger with a few updates to align it with the latest discussions in HFpEF research.
Comments on the Quality of English LanguageThe English is clear overall and the paper reads well, but there are places where shorter sentences and a bit of rephrasing would help the flow and make the arguments easier to follow. A light language edit would improve clarity and help the reader stay engaged with your findings.
Author Response
Response to Reviewer 2
Thank you for the insightful and detailed comments, which have helped improve the clarity and methodological rigor of our manuscript.
- Clarity in abstract: The abstract sets up the question well, but the phrasing around the paradox of stratification wiping out observed differences could be clearer, and reporting key numbers would help readers grasp the impact quickly:
We thank the reviewer for this insightful suggestion. To improve clarity, we have revised the abstract to better describe the apparent paradox in which sex differences observed in the overall population analysis are no longer significant after stratification. We also incorporated key numerical results to help readers quickly understand the impact: specifically, we now report hazard ratios (HRs) with 95% confidence intervals for mortality and rehospitalization, as well as event rates per 1000 patient-years and associated p-values.
- The introduction is generally solid but feels a bit standard; adding recent mechanistic insights like sex differences in microvascular dysfunction, immune aging, or fibrosis patterns would bring it up to date with where the field is moving.
We have improved the introduction to highlight this research gap more directly and provide a clearer context for our findings.
“Recent research has highlighted sex-related differences in HFpEF pathophysiology. Women show enhanced immune activation and a greater tendency toward microvascular dysfunction, which may contribute to disease development. Differences in hormonal regulation and biomarker expression have also been observed, suggesting distinct biological mechanisms between sexes that may influence prognosis and response to therapy (2).”
- The methods are well described, but defining HFpEF solely by LVEF ≥50% without natriuretic peptides or diastolic function measures risks including patients without true HFpEF, which is a known limitation in EHR studies.
We thank the reviewer for the thoughtful comment and fully agree with the limitations noted. As the SIDIAP database lacks natriuretic peptides and echocardiographic data, we sought to minimize diagnostic misclassification by including only patients with a prior hospital admission for heart failure, a strategy shown to improve diagnostic validity in EHR-based studies (Hobbs et al., Eur Heart J 2000;21:1877–1887). This clarification has been added to the limitations section.
“Additionally, the definition of HFpEF based solely on LVEF ≥50% without the availability of natriuretic peptides or diastolic function measures may have led to diagnostic misclassification. To reduce this risk, we included only patients with a documented prior hospital admission for heart failure, which has been shown to enhance diagnostic reliability in EHR-based studies (15).”
- Also, relying on backward stepwise regression may limit the robustness of your predictive models; consider discussing why this was chosen over penalized methods, and clarify how missing data were handled, as this can significantly affect results. The results are clearly presented but would benefit from explicitly reporting effect sizes with confidence intervals and providing details on model diagnostics and calibration to support your conclusions.:
We thank the reviewer for this valuable comment. Backward stepwise regression was selected for variable selection due to both methodological and practical reasons. In the absence of a pre-specified set of predictors, this approach allowed us to begin with a saturated model and retain only significant variables. While we acknowledge the value of penalized methods such as LASSO or ridge regression, we prioritized stepwise regression for its greater clinical interpretability. This rationale has been briefly added to the methods section.
Regarding missing data, we opted not to impute values and excluded incomplete cases, as missingness in real-world EHR data like SIDIAP is unlikely to be at random. Imputation could therefore introduce bias based on questionable assumptions. We recognize this may limit generalizability and have added this point to the discussion section, noting that multiple imputation will be considered in future work.
- The finding that higher LVEF is protective in women is interesting but should be discussed carefully given emerging evidence of non-linear relationships between EF and outcomes.
We thank the reviewer for this valuable observation. In response, we have added the following paragraph to the discussion. Corresponding references have also been included in the updated reference list.
“Recent evidence has questioned whether the classification thresholds for LVEF should be the same for men and women. A subanalysis of the PARAGON-HF trial (Solomon et al., Circulation. 2020;141:352–361) suggested that the prognostic cut-off for LVEF may be higher in women. However, in the same cohort, Kondo et al. did not find statistically significant sex-based differences in outcomes. Interestingly, while some studies such as Wehner et al. (Eur Heart J. 2020;41:1249–1257) report worse outcomes at higher LVEF values (i.e., supranormal EF), others show no increased risk in women with LVEF >60%. Overall, there appears to be no clear linear relationship between LVEF and prognosis when EF is above 40%. In our study, LVEF >55% was associated with a protective effect in women. Further research is needed to clarify the implications of supranormal EF on outcomes, particularly through sex-stratified analyses, as current findings remain inconclusive.”
6. The discussion aligns with your findings but could dig deeper by connecting to recent trial subgroup analyses on sex differences in HFpEF treatment response, which would increase the paper’s relevance to current debates.
We thank the reviewer for this helpful suggestion. To strengthen the discussion, we have added the following paragraph, Corresponding references have also been included in the updated reference list.
“Subgroup analyses from recent clinical trials have highlighted potential sex differences in treatment response among patients with HFpEF. In the PARAGON-HF trial, sacubitril/valsartan appeared to reduce HF hospitalizations more effectively in women, particularly those with LVEF between 45% and 57% (Solomon et al., Circulation. 2020;141:352–361). Similarly, in the TOPCAT trial, women had higher mean LVEF than men and experienced significantly lower mortality (Pitt et al., N Engl J Med. 2014;370:1383–1392). These findings suggest that sex-specific responses may influence treatment efficacy in HFpEF. In our study, no specific treatment showed a statistically significant association with outcomes, possibly reflecting differences in baseline characteristics or treatment allocation in routine clinical practice.”
7. The limitations are acknowledged but could more clearly state the potential for residual confounding and the absence of key echo parameters.
We thank the reviewer for this valuable observation. As we have previously noted (see response to point 3), we will address limitations such as potential bias, the absence of key echocardiographic parameters, and the fact that we attempted to minimize this by including only patients with a previous hospital admission.
- the conclusions are reasonable but would benefit from softer phrasing to avoid implying causality and from a clearer explanation of how sex-stratified models could improve clinical decision-making.
We thank the reviewer for the comment and ,we change de conclusion.
“Non-sex-stratified analyses in patients with HFpEF may underestimate the risk in women. Sex-stratified models could support improved clinical decision-making and adjustment of diagnostic, follow-up, and treatment protocols.”
9. The English is clear overall and the paper reads well, but there are places where shorter sentences and a bit of rephrasing would help the flow and make the arguments easier to follow. A light language edit would improve clarity and help the reader stay engaged with your findings.
We thank the reviewer for this helpful suggestion. The manuscript had already been reviewed by a native English-speaking professional with experience in scientific writing. Nevertheless, following your recommendation, we submitted the revised version for a second review by our academic English professor.
We respectfully note that no language concerns were raised by the other reviewers, but we appreciate your attention to detail and remain committed to ensuring the highest standards of scientific writing.
Reviewer 3 Report
Comments and Suggestions for Authors
Dear authors:
I found your research very interesting.
Some comments and observations:
In your introduction, you describe that evidence suggests that women can achieve therapeutic goals with lower doses than those recommended in the guidelines and that they may benefit from certain treatments (valsartan-sacubitril, spironolactone, SGLT2 inhibitors) with higher ejection fractions than men. Although it was not the objective of your research, evaluating differences in therapeutic responses generated in women compared to men would be a very important research topic.
As you describe in your discussion, in some studies, a higher LVEF has been linked to increased mortality; however, this is controversial since in your study, an LVEF >55% was a protective factor. It would be important to discuss these findings further.
I agree with you that it is important to include sex-stratified analyses in other studies to determine whether this important result obtained in your research is consistent in other cohorts from different regions.
Author Response
Response to Reviewer 3
We sincerely thank you for their valuable comments and suggestions, which have helped us to improve the quality and clarity of our manuscript. Below, we provide a point-by-point response to each of the reviewer’s comments.
- In your introduction, you describe that evidence suggests that women can achieve therapeutic goals with lower doses than those recommended in the guidelines and that they may benefit from certain treatments (valsartan-sacubitril, spironolactone, SGLT2 inhibitors) with higher ejection fractions than men. Although it was not the objective of your research, evaluating differences in therapeutic responses generated in women compared to men would be a very important research topic.
We thank the reviewer for this excellent and insightful suggestion. Although we chose not to expand the introduction to avoid redundancy, we have incorporated this valuable perspective in the discussion section. Corresponding references have also been included in the updated reference list.
“Subgroup analyses from recent clinical trials have highlighted potential sex differences in treatment response among patients with HFpEF. In the PARAGON-HF trial, sacubitril/valsartan appeared to reduce HF hospitalizations more effectively in women, particularly those with LVEF between 45% and 57% (Solomon et al., Circulation. 2020;141:352–361). Similarly, in the TOPCAT trial, women had higher mean LVEF than men and experienced significantly lower mortality (Pitt et al., N Engl J Med. 2014;370:1383–1392). These findings suggest that sex-specific responses may influence treatment efficacy in HFpEF. In our study, no specific treatment showed a statistically significant association with outcomes, possibly reflecting differences in baseline characteristics or treatment allocation in routine clinical practice.”
- As you describe in your discussion, in some studies, a higher LVEF has been linked to increased mortality; however, this is controversial since in your study, an LVEF >55% was a protective factor. It would be important to discuss these findings further
We thank the reviewer for the comment and add the following paragraph to the discussion. We added a paragraph in discussion about it. Corresponding references have also been included in the updated reference list.
“Recent evidence has questioned whether the classification thresholds for LVEF should be the same for men and women. A subanalysis of the PARAGON-HF trial (Solomon et al., Circulation. 2020;141:352–361) suggested that the prognostic cut-off for LVEF may be higher in women. However, in the same cohort, Kondo et al. did not find statistically significant sex-based differences in outcomes. Interestingly, while some studies such as Wehner et al. (Eur Heart J. 2020;41:1249–1257) report worse outcomes at higher LVEF values (i.e., supranormal EF), others show no increased risk in women with LVEF >60%. Overall, there appears to be no clear linear relationship between LVEF and prognosis when EF is above 40%. In our study, LVEF >55% was associated with a protective effect in women. Further research is needed to clarify the implications of supranormal EF on outcomes, particularly through sex-stratified analyses, as current findings remain inconclusive.”